# How Bad is Training on Synthetic Data? A Statistical Analysis of Language Model Collapse

**Mohamed El Amine Seddik**
Technology Innovation Institute
Abu Dhabi, UAE
mohamed.seddik@tii.ae

**Suei-Wen Chen**
NYU Abu Dhabi
Abu Dhabi, UAE
swc435@nyu.edu

**Soufiane Hayou**
Simons Institute
Berkeley, USA
hayou@berkeley.edu

**Pierre Youssef**
NYU Abu Dhabi
Abu Dhabi, UAE
yp27@nyu.edu

**Merouane Debbah**
Khalifa University of Science and Technology
Abu Dhabi, UAE
merouane.debbah@ku.ac.ae

## Abstract

The phenomenon of model collapse, introduced in (Shumailov et al., 2023), refers to the deterioration in performance that occurs when new models are trained on synthetic data generated from previously trained models. This recursive training loop makes the tails of the original distribution disappear, thereby making future-generation models forget about the initial (real) distribution. With the aim of rigorously understanding model collapse in language models, we consider in this paper a statistical model that allows us to characterize the impact of various recursive training scenarios. Specifically, we demonstrate that model collapse cannot be avoided when training solely on synthetic data. However, when mixing both real and synthetic data, we provide an estimate of a maximal amount of synthetic data below which model collapse can eventually be avoided. Our theoretical conclusions are further supported by empirical validations.

## 1 Introduction

The large-scale adoption of large language models (e.g. ChatGPT (OpenAI, 2024)) will inevitably lead to enormous amounts of synthetic (generated) data "polluting" the original human-created web data. Since language models are trained on web data, this raises some concerns about the impact of this synthetic data on the next generations of LLMs. One can think of a train-generate loop where models from the current generation generate data that contaminate existing web data, and the next generation models are trained on this contaminated data. This loop was studied in several works (Shumailov et al., 2023; Alemohammad et al., 2023; Briesch et al., 2023) where the authors conclude that synthetic data generally hurts performance as the number of generate-train increases, and that (naturally) the impact on model performance is linked to the amount of real data in the training set. A particular phenomenon, coined *model collapse* (Shumailov et al., 2023), refers to the model's tendency to produce limited or repetitive outputs making recursive training on such outputs forget about the tails of the original underlying distribution of real data. This was further studied in (Guo et al., 2023) where the authors show that recursive training on synthetic data leads to a "self-consuming" loop that affects linguistic diversity.

Intuitively, model collapse is a result of the distribution shift that occurs when training generative models recursively on synthetic data from previous generation models. Shumailov et al. (2023) have discussed two main sources of model collapse; *1) statistical approximation error:* which is inherently related to the fact that generative models are trained on a *finite* number of samples, and therefore it is impossible that the learned model captures all the information about the original distribution. *2) functional approximation error:* which results from

the fact that the models are insufficiently expressive in real implementations, even if neural networks are known to be universal functional approximators from a theoretical standpoint. The authors provide further theoretical intuition to characterize the effect of these approximation errors, relying on simple mathematical models such as single-dimensional Gaussian distribution.

In this paper, we aim to provide a rigorous theoretical framework to understand the effects of recursive training with synthetic data. In particular, we focus on the statistical approximation error and introduce a simple next-token-prediction language model to characterize model collapse. Our model allows us to gain insights into the behaviour of the self-consuming train-generate loop leading to model collapse. From a theoretical standpoint, we consider two main recursive training scenarios:

- *Fully Synthetic:* Training with data sampled from the previous generation model.

- *Partially Synthetic:* Training with a mixture of data sampled from the previous generation model and original data.

We demonstrate that model collapse always occurs in the first scenario and characterize the rates at which it occurs. Furthermore, in the second scenario, we provide an upper bound on the sample size of generated data below which model collapse can eventually be attenuated. Our results are further confirmed through simulations of general scenarios with the introduced statistical model, as well as with realistic GPT2-style language models on real data. Our findings suggest that the amount of generated data should be considerably smaller compared to the original data to avoid model collapse.

**Related work:**  With the adoption of generative Large Language and Vision models, the amount of synthetic data on the web is growing at an unprecedented rate – see for example (del Rio-Chanona et al., 2023) where the authors conducted an empirical study of the amount of synthetic data via activity monitoring on Stack Overflow, and (Alemohammad et al., 2023) where the authors showed that a dataset used to train Stable Diffusion contains synthetic data (Schuhmann et al., 2022). In fact, practitioners are willingly using synthetic data to train next-generation models (Ben Allal et al., 2024; Gunasekar et al., 2023; Chen et al., 2024).

As we mentioned above, several works studied the recursive training loop where next-generation models are trained on synthetic data generated from previous-generation models. Shumailov et al. (2023) studied model collapse, a phenomenon that occurs in recursive training where the quality of model outputs tend to degrade by becoming e.g. repetitive. Similar phenomena were studied in (Alemohammad et al., 2023) where the authors call it Model Autophagy Disorder (MAD). Another empirical study by (Briesch et al., 2023) studied this same Self-Consuming phenomenon and observed that the degeneracy rate (naturally) depends on the number of fresh data in the training sample. In the same direction, several works showed that incorporating synthetic data in the training can hurt the performance of trained diffusion models (Bohacek & Farid, 2023; Martínez et al., 2023a;b).

Only a few works have tackled this question from a theoretical perspective. Shumailov et al. (2023) considered a simple recursive Gaussian distribution to provide an intuition as to why model collapse occurs, but no training is considered in that work. In (Fu et al., 2024), authors studied recursive training of diffusion models (generally used to learn distributions over images) and obtained an upper bound on the total variation distance between the distribution of the original data and that of the synthetic data after $T$ generations. Dohmatob et al. (2024b) studied the impact of synthetic data on scaling laws and, in a simple setting of linear regression, Dohmatob et al. (2024a) studied the behaviour of the test error of different generations and showed that a linear dependency of the degradation on generation number.

In this work, we are interested in characterizing the *distribution shift* in synthetic data generated with a language model, as the number of generations increases. We consider a linear Softmax classifier for next-token prediction and study the distribution of the learned conditional probabilities as the number of generations increases. The closest work to ours is (Fu et al., 2024) where the authors study the distribution of synthetic data generated by a

diffusion model instead, whereas the statistical model we are considering is closer in spirit to the realm of language models.

The remainder of the paper is organized as follows. Section 2 presents our theoretical setup where we introduce our statistical model and the considered recursive training scenarios. Our main theoretical results are presented in Section 3. We further present some experiments in Section 4 to support our findings. Finally, Section 5 concludes the paper.

# 2 Theoretical Setup

## 2.1 Statistical Language Model

We consider a language model of vocabulary size $s$, context length denoted by $\ell$ and we further denote by $c$ the number of possible contexts which is at most $s^\ell$. We suppose that the language data is generated from some unknown conditional probabilities given the contexts. That is, the probability of the next token being $k \in [s] := \{1, ..., s\}$ given some context $\boldsymbol{j} = (j_1, \ldots, j_\ell) \in [c]$ is denoted by

$$p_{jk} := \mathbb{P}\{Y = k \mid X = \boldsymbol{j}\},$$

where $X$ and $Y$ denote discrete random variables representing a context and the next-token respectively. In practice, we do not have access to the true conditional probabilities $p_{jk}$ but rather a (large) corpus sampled according to $p_{jk}$. In other words, we are given a dataset $\{(\boldsymbol{x}_l, \boldsymbol{y}_l)\}_{l \in [M]}$ of $M$ samples of contexts and next-token pairs represented by $\boldsymbol{x}_l \in \{\boldsymbol{e}_1, \ldots, \boldsymbol{e}_c\}$ and $\boldsymbol{y}_l \in \{\boldsymbol{e}_1, \ldots, \boldsymbol{e}_s\}$ where $\boldsymbol{e}_i$'s denote the canonical vectors.

Given this dataset, we consider approximating the underlying conditional probabilities via the Softmax classifier, which entails minimizing the categorical cross-entropy loss function:

$$\underset{\mathbf{W}=[\boldsymbol{w}_1, \ldots, \boldsymbol{w}_s] \in \mathbb{R}^{c \times s}}{\arg\min} \; -\frac{1}{M} \sum_{l=1}^{M} \boldsymbol{y}_l^\top \log \sigma\left(\mathbf{W}^\top \boldsymbol{x}_l\right), \tag{1}$$

where $\sigma(\boldsymbol{v}) = \frac{\exp(v)}{\sum_{k=1}^{s} \exp(v_k)}$ is the Softmax function and the functions exp and log are applied entry-wise. Note that in current state-of-the-art language models, the $\boldsymbol{x}_l$'s are context representations computed via transformer models, whereas, in our setting, we choose to work with one-hot embeddings as representations for tractable theoretical analysis. Solving the above objective (see Appendix 8.1) yields the estimated conditional probability $\hat{p}_{jk}$ of $p_{jk}$ which expresses as the following empirical mean:

$$\hat{p}_{jk} = \frac{\exp(\hat{\boldsymbol{w}}_k^\top \boldsymbol{e}_j)}{\sum_{i=1}^{s} \exp(\hat{\boldsymbol{w}}_i^\top \boldsymbol{e}_j)} = \frac{1}{|\mathcal{C}_j|} \sum_{l \in \mathcal{C}_j} y_{lk} \quad \text{with} \quad \mathcal{C}_j = \{l \in [n] \mid \boldsymbol{x}_l = \boldsymbol{e}_j\}. \tag{2}$$

The estimated conditional probabilities $\hat{p}_{jk}$ are the result of training on the original data. These conditional probabilities can be used to generate new synthetic data, which can be used (with or without additional fresh data from the original dataset) to train the next-generation model. In this paper, we are interested in characterizing the behaviour of $\hat{p}_{jk}$ in this recursive training loop which we will formally define in the next section. Hereafter, without loss of generality, we consider a fixed context $\boldsymbol{j}$ with $N := |\mathcal{C}_j|$ training samples $\{(\boldsymbol{e}_j, \boldsymbol{y}_l)\}_l$ of context-next-token pairs, where $\boldsymbol{y}_l \in \mathbb{R}^s$ are independent multinomial random variables with one trial and parameter

$$\boldsymbol{p} = (p_1, \ldots, p_s) := (p_{j1}, \ldots, p_{js}) \in [0, 1]^s. \tag{3}$$

From (2), we notice that $\hat{p}_{jk}$ is an estimate of $p_{jk}$ and we further denote

$$\boldsymbol{p}^{(1)} = (\hat{p}_1, \ldots, \hat{p}_s) := (\hat{p}_{j1}, \ldots, \hat{p}_{js}) \in [0, 1]^s. \tag{4}$$

As such, $\boldsymbol{p}^{(0)} := \boldsymbol{p}$ corresponds to the ground-truth distribution whereas $\boldsymbol{p}^{(1)}$ denotes the first-generation model trained on the real data $\{(\boldsymbol{e}_j, \boldsymbol{y}_l)\}_{l \in \mathcal{C}_j}$.

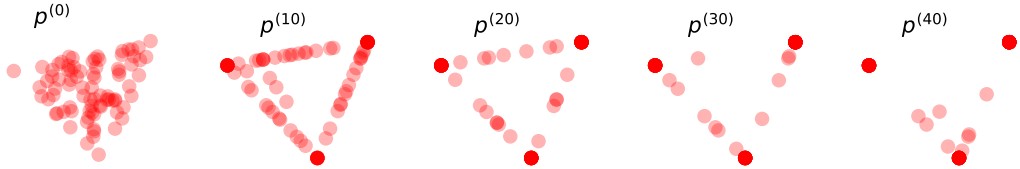

Figure 1: Evolution of $p^{(m)}$ in the *Fully Synthetic* setting for vocabulary size $s = 3$, context length $\ell = 4$, total contexts $c = s^\ell = 81$ and sample size $n = 1000$. The initial distribution $p^{(0)}$ is some random distribution over tokens. The trained conditional distributions converge towards Dirac measures over generations illustrating *total collapse* in Definition 1.

## 2.2 Recursive Training

In this section, we introduce the notations for recursive training. At generation $m \geq 1$, suppose that we have samples $\{y_l^{(t)}\}_l$ generated by some past models $p^{(t)}$ for $t \in \{0, \ldots, m-1\}$ respectively. As such, similarly to (2), the model at generation $m$ expresses as

$$p^{(m)} := \frac{1}{n^{(m)}} \sum_{t=0}^{m-1} \sum_{l=1}^{n_t^{(m)}} y_l^{(t)}. \tag{5}$$

In other words, $p^{(m)}$ is obtained by training the model on a mixture of real and synthetic data generated from previously trained models. Here, $n_t^{(m)}$ stands for the number of samples used to train the $m$-th generation model that are generated by model $p^{(t)}$, and $n^{(m)} = n_0^{(m)} + \cdots + n_{m-1}^{(m)}$ is the total number of training samples used to train $p^{(m)}$. Note that we do not generate samples from $p = p^{(0)}$ but rather we are given a corpus $\{y_l^{(0)}\}_l$ which represent original data.

Note that by definition all the models $p^{(m)}$ are unbiased, i.e. $\mathbb{E}p^{(m)} = p$, which means that they recover the original distribution in case of infinite sample sizes. However, in practical settings the sample sizes are finite, and therefore $p^{(m)}$ may deviate from $p$ since its variance can be large for small values of $n_t^{(m)}$. In the remainder of the paper, we present analytical results to rigorously quantify the impact of sample sizes $n_t^{(m)}$ on the variance of $p^{(m)}$ and how they affect the learned distribution to eventually cause *model collapse* (Shumailov et al., 2023), a degenerative process affecting future generation models by either losing information about the tails of the initial distribution or inducing distribution shifts over generations.

Specifically, we aim to quantify the rate of such deterioration, and to this end, we introduce a stricter version of model collapse that we call *total collapse* and define as follows.

**Definition 1** (Total Collapse). *We say that total collapse occurs in the recursive training process if $p^{(m)}$ converges to some Dirac mass $\delta_i$ for some token $i \in [s]$ as $m$ grows.*

Total collapse refers to the case where, under recursive training, the trained model $p^{(m)}$ completely loses information about the original distribution $p$ over generations, leading to poor linguistic diversity.[1] This phenomenon is illustrated in Figure 1 where we see convergence towards the vertices of the probability simplex, i.e. Dirac measures, over generations. With this definition of total collapse, we provide a quantitative analysis of two cases of recursive training:

---

[1]Note that here we define total collapse with respect to a single pre-fixed context. This can be generalized to the event where total collapse occurs for all contexts. However, the theoretical analysis in this case would require more refined treatment of several statistical quantities to obtain uniform bounds over contexts. We leave this for future work.

- *Fully Synthetic:* Training with synthetic data from the last model. Each generation $\boldsymbol{p}^{(m)}$ is trained only on data generated by the previous model $\boldsymbol{p}^{(m-1)}$. More precisely, for some fixed $n \in \mathbb{N}$ we let $n_t^{(m)} = n \cdot \mathbb{1}\{t = m-1\}$ for all $m \geq 1$ and $0 \leq t < m$.
- *Partially Synthetic:* Training with a mixture of real and synthetic data from the last model. Each generation $\boldsymbol{p}^{(m)}$ is trained on a mixture of real data and synthetic data generated by the previous model $\boldsymbol{p}^{(m-1)}$. More precisely, for some fixed $n \in \mathbb{N}$ we let $n_0^{(m)} = N$, $n_{m-1}^{(m)} = n$ and $n_t^{(m)} = 0$ for all $m \geq 2$ and $0 < t < m$, and $n_0^{(1)} = N$.

*Fully Synthetic* corresponds to the theoretical setting considered in (Shumailov et al., 2023). However, in that paper, only Gaussian distribution was analyzed instead of discrete distributions over all possible tokens as language models entail. We point out that this setting is unlikely to happen in real-world applications but serves as the worst-case scenario.

On the other hand, the *Partially Synthetic* setting is more realistic for future generation models as it corresponds to training on a mixture of real and synthetic data. We consider this setting to assess whether it is possible to avoid collapse by having a fraction of the original data in the training mixture. We answer this question positively in the next section. Moreover, we show through simulations in Section 4 that conclusions from our theoretical analysis on both *Fully Synthetic* and *Partially Synthetic* hold beyond these simple settings, such as training on a mixture of all generations or even using realistic transformer models.

# 3 Main Results

To investigate the model collapse phenomenon and the rate at which it occurs, we define the following statistical quantities that capture the randomness of $\boldsymbol{p}^{(m)}$:

$$\|\boldsymbol{p}^{(m)}\|_\infty := \max_{i \in [s]} p_i^{(m)}, \quad \sigma_m := \|\boldsymbol{p}^{(m)}\|_2^2 = \sum_{i=1}^{s} p_i^{(m)^2} \quad \text{and} \quad S_m := \mathbb{E}[\sigma_m].$$

These quantities measure how far away $\boldsymbol{p}^{(m)}$ is from some Dirac mass (Total Collapse, Definition 1). Since the maximum value of all three quantities is 1, the closer they are to 1, the closer $\boldsymbol{p}^{(m)}$ is to some Dirac mass, and the less diverse $\boldsymbol{p}^{(m)}$ is as a language model. As such, $\|\boldsymbol{p}^{(m)}\|_\infty$ or $\sigma_m$ being equal to 1 is equivalent to $\boldsymbol{p}^{(m)}$ being a Dirac mass which reflects total collapse. To quantify the distribution shift incurred by the aforementioned recursive training scenarios, we further consider the 1-norm[2] between two distributions $\mu, \nu \in \mathbb{R}^s$ defined by $\|\mu - \nu\|_1 := \sum_{i=1}^{s} |\mu(i) - \nu(i)|$.

We assume that the initial distribution $\boldsymbol{p}^{(0)}$ is *nontrivial*, specifically $S_0 < 1$ and $\|\boldsymbol{p}^{(0)}\|_\infty < 1$. Under this assumption, we provide results on the rate of total collapse and further quantify distribution shift under recursive training. All the proofs are presented in Appendix 8.

## 3.1 *Fully Synthetic:* Training on synthetic data

We start by describing the recursive training process in which only synthetic data from the last generation model are used. This process can be viewed as a Markov chain on the set of probability measures $\Delta^{s-1} \cap \frac{1}{n}\mathbb{N}^s$ on $[s]$ with denominator $n$, where

$$\Delta^{s-1} := \{\boldsymbol{v} \in \mathbb{R}^s : v_1 + v_2 + \cdots + v_s = 1, v_i \geq 0 \text{ for all } i\}$$

---

[2]We point out that the 1-norm is twice the total variation distance $\|\mu - \nu\|_{\text{TV}}$, which is another commonly used metric for probability distributions and was considered by Fu et al. (2024) for studying model collapse in the case of diffusion models.

is the probability simplex. Since the probability of reaching $\delta_i$ is positive provided $p_i > 0$, this random walk which starts at $\boldsymbol{p}^{(0)}$ has absorbing states $\{\delta_i : i \in [s] \text{ s.t. } p_i > 0\}$. As a result, the random walk converges to one of the absorbing states almost surely (Kemeny et al., 1969), which means that total collapse is bound to happen in the *Fully Synthetic* setting.

To characterize the rate at which total collapse occurs, let us denote by $T := \inf\{t \in \mathbb{N} : \|\boldsymbol{p}^{(t)}\|_\infty = 1\} \geq 1$ the random time at which the model first becomes a Dirac mass, and let $\rho_m := \mathbb{P}\left(\|\boldsymbol{p}^{(m)}\|_\infty = 1\right)$ denote the probability that the $m$-th generation has collapsed. Our first result, presented in Theorem 1, provides the rate of convergence via $S_m$, $\rho_m$ and $\mathbb{E}[T]$.

**Theorem 1** (Control on Total Collapse). *Consider the Fully Synthetic setting and let $\tilde{s} := |supp(\boldsymbol{p})|$ denote the support size of $\boldsymbol{p}$, namely $\tilde{s} := |\{i \in [s] : p_i > 0\}|$.*

1. *The expected sum of squared probabilities $S_m$ is given by*

$$S_m = 1 - \left(1 - \frac{1}{n}\right)^m (1 - S_0). \tag{6}$$

2. *The probability $\rho_m$ that total collapse has occurred by generation m satisfies*

$$1 - n(1 - S_0)(1 - 1/n)^m \leq \rho_m \leq 1 - \frac{1 - S_0}{1 - 1/\tilde{s}}(1 - 1/n)^m. \tag{7}$$

3. *The generation T at which total collapse happens satisfies*

$$1 + \frac{1 - S_0}{1 - 1/\tilde{s}}(n - 1) \leq \mathbb{E}[T] \leq 1 + (1 - S_0)n(n - 1). \tag{8}$$

In essence, Theorem 1 describes the behavior of $\boldsymbol{p}^{(m)}$ as a function of the model generation $m$, the sample size $n$, and the dispersion of the initial distribution $S_0$. Specifically, we draw the following observations:

- Effect of the number of generations $m$: As $m$ increases, $S_m$ and $\rho_m$ tend to 1 as per (6) and (7), making total collapse increasingly more likely. Note also that this dependence is exponential in $m$, making total collapse in this case relatively fast.

- Effect of "synthetic" sample size $n$: The smaller $n$ is, the more likely $\boldsymbol{p}^{(m)}$ is to have collapsed for a given generation $m$ as per the bounds on $\rho_m$ in (7), and the faster total collapse is expected to happen as suggested by (8). The dependence in this case is polynomial in $n$.

- Effect of $S_0$: Larger values of $S_0$ correspond to faster collapse as suggested by (8); namely, starting from an original distribution that is not diverse enough speeds up total collapse. The upper and lower bounds on $\mathbb{E}[T]$ are both linear in $(1 - S_0)$.

We point out that when the number of contexts $c$ is large, the number of samples $n$ per context would be fairly small, which suggests that total collapse, given a single context, is expected to happen fairly quickly. The bounds on $\mathbb{E}[T]$ state that the expected collapse time is at least of order $n$ and at most of order $n^2$, though experiments (see Figure 2) suggest that the upper bound is not sharp and should be close to $\mathcal{O}(n)$ instead. On another note, we characterize below in Proposition 1 the limiting distribution as $m$ gets to infinity, which demonstrates the direction of total collapse.

**Proposition 1.** *In the Fully Synthetic case, we have $\mathbb{P}\left(\lim_{m \to \infty} \boldsymbol{p}^{(m)} = \delta_i\right) = p_i$ for all $i \in [s]$.*

Proposition 1 describes the limiting distribution when total collapse occurs in terms of the initial probabilities $p_i$ over tokens. Specifically, the resulting Dirac mass $\delta_i$ is likely to be supported on some token $i$ with high initial probability $p_i$. This formally supports the description of *early* and *late* model collapse in Shumailov et al. (2023): In the early phase of recursive training, the tails of the original distribution disappear because the probability

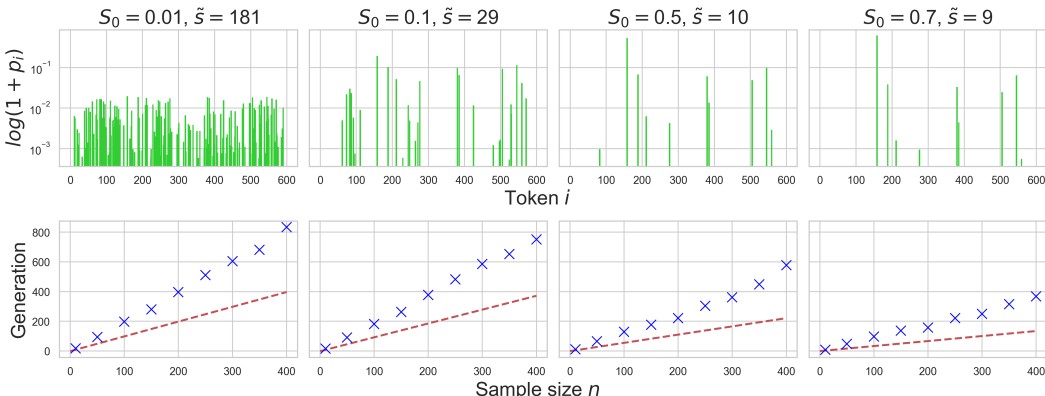

Figure 2: **Fully Synthetic case for different initial distribution** $p^{(0)}$ **.** Total collapse time is plotted as a function of the initial distribution $p$ and the sample size $n$. **(Top)** The initial distribution $p$ with different values of $S_0$ and support size $\tilde{s}$. The $x$-axis represents tokens $i \in \{1, 2, \ldots, 600\}$ while the $y$-axis represents the probabilities in log scale. **(Bottom)** Each cross represents the average total collapse time over 100 runs for a particular sample sizes $n \in \{10, 50, 100, 150, \ldots, 400\}$. The red dashed line depicts the lower bound on $\mathbb{E}T$ given by (8).

$p_j^{(m)}$ of outputting unlikely tokens $j$ (those $j$'s for which $p_j$ is small) will decrease as $p^{(m)}$ converges to some Dirac mass $\delta_i$. After many generations, all but one $p_j^{(m)}$ will go to 0, exhibiting late model collapse where all the randomness of the original distribution is lost.

## 3.2 *Partially Synthetic:* **Handling model collapse with real data**

As described in the previous section, total collapse is unavoidable when training solely on synthetic data. In this section, we consider the case in which real data is incorporated at each generation. In this case, $p^{(m)}$ can be simply seen as a weighted average of $p^{(1)}$ and an estimate of $p^{(m-1)}$ with $n$ samples. We also assume that $p^{(1)}$ is nontrivial, thereby implying that, on average, $p^{(m)}$ will not be a Dirac mass. In what follows, we quantify the variability in the data distribution across different generations, as well as the distribution drift $\|p^{(m)} - p^{(1)}\|_1$ from the first-generation model. We particularly show that collapse can be avoided if enough real data is injected into the recursive training process.

**Theorem 2** (Model Variation). *Consider the Partially Synthetic setting. For $m \geq 1$ we have*

$$S_{m+1} = \frac{\frac{1}{N}\left[1 + 2\alpha - \left(1 - \frac{1}{N}\right)\alpha\beta^m\right]}{1 + (1 + 1/N)\alpha} + \frac{(1 - \frac{1}{N})S_0}{1 + (1 + 1/N)\alpha}\left[1 + \alpha - \frac{\alpha\beta^m}{N}\right], \qquad (9)$$

*where $\alpha := \frac{n}{N+n}$ and $\beta := \alpha\left[(1 + \frac{1}{N})\alpha - \frac{1}{N}\right]$.*

Theorem 2 provides a control of the variance $S_m$ in the *Partially Synthetic* setting. Essentially, when $n \ll N$, we have $S_m \approx \frac{1}{N} + (1 - \frac{1}{N})S_0 \approx S_0$, which is not surprising since the training data for each generation model are dominated by real data in this case. However, even when the number of synthetic data is much larger than the original dataset (i.e. $n \gg N$), we have $\alpha \approx 1 \approx \beta$ and hence $S_{m+1} \approx 1/N + (2 + 1/N)^{-1}(1 - 1/N)(2 - 1/N)S_0$, which approaches $S_0$ for large $N$.

To further refine our analysis, we present a result that directly controls the deviation $\mathbb{E}\|p^{(m)} - p^{(1)}\|_1$ between the conditional distributions from first and $m$-th generations. This allows us to have a quantitative control over the distribution shift. When $n$ is sufficiently small, we have a sharper control over this deviation by exploiting the concentration results

from (Mardia et al., 2020). Essentially, this allows us to estimate the maximum number of synthetic samples $n$ to ensure that the distribution $\boldsymbol{p}^{(m)}$ stays close to $\boldsymbol{p}^{(1)}$.

**Theorem 3** (Model Deviation). *Consider the Partially Synthetic setting and define*

$$G_n(s) := \begin{cases} C_1 s e^{\frac{C_0 n}{2e}} & \text{if} \quad \frac{C_0}{e}n + 2 \leq s; \\ C_1 s \left(\frac{C_0 n}{s}\right)^{s/2} & \text{if} \quad \frac{C_0}{4}n + 2 \leq s < \frac{C_0}{e}n + 2; \\ (2^s - 2) & \text{if} \quad s < \frac{C_0}{4}n + 2, \end{cases} \tag{10}$$

*where $C_0 = \frac{e^3}{2\pi} \approx 3.19$ and $C_1 = \frac{6e}{\pi^{3/2}} \approx 2.93$, and $s$ is the vocabulary size. Then, for $m \geq 2$,*

$$\mathbb{E}\|\boldsymbol{p}^{(m)} - \boldsymbol{p}^{(1)}\|_1 < \frac{1}{N}\sqrt{\frac{\pi n}{2}} G_n(s).$$

We point out that the upper bound on the deviation $\mathbb{E}\|\boldsymbol{p}^{(m)} - \boldsymbol{p}^{(1)}\|_1$ is independent of the generation $m$. Since the deviation from $\boldsymbol{p}^{(0)}$ to $\boldsymbol{p}^{(1)}$ is inevitable and independent of $n$, we give the result in $\mathbb{E}\|\boldsymbol{p}^{(m)} - \boldsymbol{p}^{(1)}\|_1$, from which $\mathbb{E}\|\boldsymbol{p}^{(m)} - \boldsymbol{p}^{(0)}\|_1$ can be estimated by the triangle inequality. Theorem 3 allows us to estimate the maximum number of synthetic samples $n$ that can be used if we want $\boldsymbol{p}^{(m)}$ to stay $\epsilon$-close to $\boldsymbol{p}^{(1)}$ in $L^1$ norm. For example, when $C_0 n/e + 2 \leq s$, for any $\epsilon > 0$ we can take

$$n \leq 2\pi e^{-2} \min\left[s - 2, \log\left(\frac{\sqrt{2}\pi N\epsilon}{6es}\right)\right] \tag{11}$$

in order for $\mathbb{E}\|\boldsymbol{p}^{(m)} - \boldsymbol{p}^{(1)}\|_1$ to be less than $\epsilon$. In other words, to ensure small $L^1$ deviation $n$ should be taken to be logarithmic in the ratio $N\epsilon/s$, which highlights that the amount of synthetic data should be exponentially smaller compared to real data in order to ensure that $\boldsymbol{p}^{(m)}$ remains close to $\boldsymbol{p}^{(1)}$. In Appendix 9, we provide experiments showing the effect of the sample size $n$ and the initial distribution $\boldsymbol{p}$. In Figure 5, we show that if we fix the initial distribution and increase the amount of synthetic data $n$, the dispersion $\sigma_m$ stays relatively constant while the distribution drift $\|\boldsymbol{p}^{(m)} - \boldsymbol{p}^{(1)}\|_1$ increases. In contrast, when we fix $n$ and increase the values of $S_0$, $\sigma_m$ increases but $\|\boldsymbol{p}^{(m)} - \boldsymbol{p}^{(1)}\|_1$ actually decreases as depicted in Figure 6. This behavior can be explained by Theorem 4 in Appendix 8, which is more general than Theorem 3 in the sense that it captures the dependence of $\mathbb{E}\|\boldsymbol{p}^{(m)} - \boldsymbol{p}^{(1)}\|_1$ on the randomness of $\boldsymbol{p}$ and hence provides a sharper bound on the expected deviation. We provide additional simulations in Appendix 9.2 to account more general scenarios.

# 4 Experiments

To support our findings in a realistic setting, we conduct experiments with a decoder-only generative model for text generation. For this model, we consider the aforementioned *Fully Synthetic* and *Partially Synthetic* settings using the model parameters in Appendix 9.1. We consider a simple character-level tokenizer using the tiny Shakespeare dataset[3] yielding a vocabulary of size $s = 65$. We first train a model for 2000 iterations on this dataset and consider the trained model as the ground-truth distribution $\boldsymbol{p}^{(0)}$. The next-generation models are trained recursively using the exact same architecture and training parameters. This setting allows us to reduce the effect of functional approximation error thereby focusing only on the effect of statistical approximation error.

The results of these experiments are summarized in Figure 3. The top left plot therein depicts the deviation $\|\boldsymbol{p}^{(m)} - \boldsymbol{p}^{(1)}\|_1$ which is averaged over 300 random contexts generated by

---

[3]https://raw.githubusercontent.com/karpathy/char-rnn/master/data/tinyshakespeare/input.txt

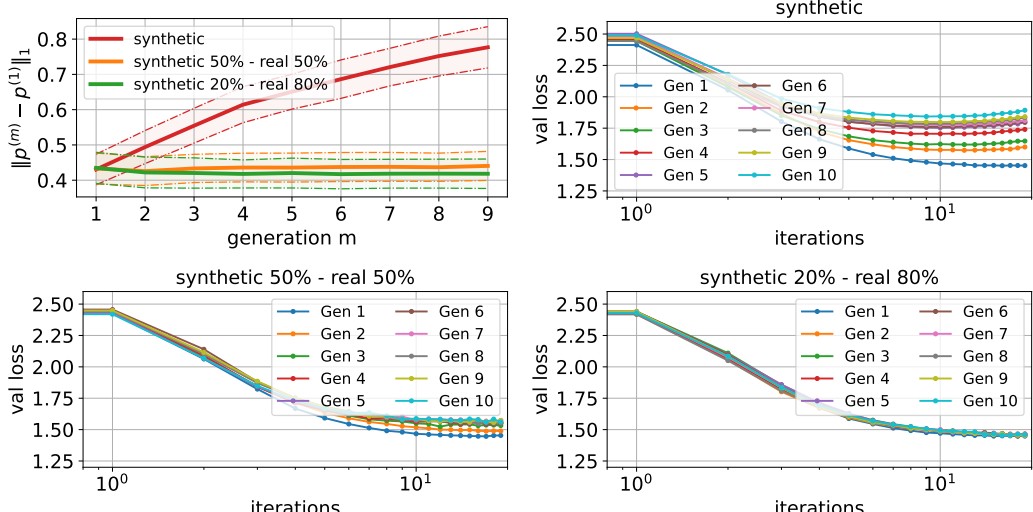

Figure 3: Experiments with a GPT2-type generative model. The top left plot depicts the deviation $\|\boldsymbol{p}^{(m)} - \boldsymbol{p}^{(1)}\|_1$ varying the generation model $m$ for synthetic data and a mixture of real and synthetic data. The three other plots show the behavior of the validation loss over generations. Essentially, training solely on synthetic data causes model collapse and affects the usual scaling laws (Dohmatob et al., 2024a).

the ground-truth generative model $\boldsymbol{p}^{(0)}$. From this plot, we clearly see that $\|\boldsymbol{p}^{(m)} - \boldsymbol{p}^{(1)}\|_1$ diverges over generations in the *Fully Synthetic* setting, while mixing with real data ensures stability over generations.

The effect of synthetic data is further noticed in the validation loss of the next-generation models where we see an overfitting effect in the *Fully Synthetic* setting. We point out that this effect might also be associated with the functional approximation error and was rigorously studied by Dohmatob et al. (2024a) in the case of linear regression, where the authors have shown that synthetic data affect the usual scaling laws. We believe that similar conclusions can be obtained with our statistical model by incorporating the functional approximation error, this can be achieved for instance by supposing that the context embeddings $\boldsymbol{e}_i$'s are high-dimensional Gaussian vectors instead of canonical vectors, thereby introducing the embedding dimension as a parameter controlling model complexity.

# 5  Discussions & Conclusion

In this paper, we studied model collapse in language models through a simple statistical model. We provided theoretical analysis when training with only synthetic data and when adding real data from the original distribution. Our results demonstrate that model collapse always happens when the model is training solely on synthetic data, whereas controlling deviation from the initial distribution requires careful choice of the amount of synthetic data to inject in the training set. We also provided experiments showing that these findings extend beyond the simple theoretical settings.

Our current results describe only the statistical approximation error since all generation models are unbiased in our theoretical framework. However, as we discussed in the previous section, this framework can be extended to account for the functional approximation error by considering high-dimension Gaussian vectors as context embeddings instead of canonical vectors. Another possible extension is to study the effect of in-context learning (Wu et al., 2023) on model collapse, which is a key feature of transformer-based models.

Despite the simple setting of our current investigation, we believe that it lays the groundwork for better understanding and mitigation of model collapse in language models, thereby opening the way for the development of more general theoretical frameworks to study next-generation language models dynamics.

# 6 Limitations

Our analysis primarily focuses on a simplified recursive training setup, which may not fully represent current large language model training practices. The theoretical results are derived for a basic language model, and extending the analysis to more complex architectures like transformers remains challenging. Our empirical validations use relatively small-scale experiments due to computational constraints, and the generalizability of these findings to large-scale language models requires further investigation. While we explore more realistic settings in the appendix, such as "Most Recent Models" and "Randomly Sampled Data", these still represent idealizations of real-world scenarios. Additionally, our treatment of synthetic data generation does not account for various post-processing techniques that may be employed in practice. Future work should address these limitations by expanding the theoretical framework to more complex model architectures, conducting larger-scale experiments, and considering a broader range of synthetic data generation and utilization approaches.

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

# 7   Technical Lemma

For completeness, we include the concentration results from (Mardia et al., 2020) which we used to prove Theorem 3. For a probability distribution $p$ on $[s]$ define

$$\pi_{\boldsymbol{p}} := \max_{A \subseteq [s]} \min\{\boldsymbol{p}(A), 1 - \boldsymbol{p}(A)\} \tag{12}$$

and notice that $\pi_{\boldsymbol{p}} \leq \frac{1}{2}$. Define the function $\varphi : [0, 1/2) \to \mathbb{R}$ by

$$\varphi(x) := \frac{1}{1 - 2x} \log \frac{1 - x}{x}$$

and extend $\varphi$ by continuity to $\varphi(1/2) := 2$. Observe that $\varphi$ is strictly decreasing and $2 \leq \varphi(x) < \infty$. The following Lemma concerns the concentration of empirical distribution which captures the dependence on the sample size $n$, dimension $s$ as well as the structure of the underlying probability distribution via $\varphi(\pi_{\boldsymbol{p}})$.

**Lemma 1.** *Let $p$ be a probability distribution on $[s]$ and $\hat{p}$ be the associated empirical distribution obtained from $n$ samples. Then for $\epsilon > 0$ we have*

$$\mathbb{P}\left(\|\hat{\boldsymbol{p}} - \boldsymbol{p}\|_1 \geq \epsilon\right) \leq \exp\left(-\frac{n\varphi(\pi_{\boldsymbol{p}})\epsilon^2}{4}\right) G_n(s),$$

*where $C_0 = \frac{e^3}{2\pi}$, $C_1 = \frac{6e}{\pi^{3/2}}$ and*

$$G_n(s) = \begin{cases} C_1 s \left(C_0 n/s\right)^{s/2} & \text{if } \frac{C_0 n}{4} + 2 \leq s \leq \frac{C_0 n}{e} + 2; \\ C_1 s e^{\frac{C_0 n}{2e}} & \text{if } \frac{C_0 n}{e} + 2 \leq s; \\ (2^s - 2) & \text{if } s \leq \frac{C_0 n}{4} + 2. \end{cases} \tag{13}$$

The upper bound

$$\mathbb{P}(\|\hat{\boldsymbol{p}} - \boldsymbol{p}\|_1) \leq (2^s - 2)\exp\left(-\frac{n\varphi(\pi_{\boldsymbol{p}})\epsilon^2}{4}\right)$$

in fact holds for all values of $s$ and $n$ (Weissman et al., 2003), but this bound is very poor when $s/n$ is large, which is commonly the case for language models. Lemma 1 provides an improvement in the regime of small sample size $n \lesssim s$.

# 8 General Results and Proofs

## 8.1 Derivation of Formula (2)

*Proof.* The categorical cross-entropy loss reads as

$$\mathcal{L}(\boldsymbol{w}_1, \ldots, \boldsymbol{w}_s) = -\frac{1}{M}\sum_{i=1}^{M}\sum_{k=1}^{s} y_{ik}\log\left(\frac{\exp(\boldsymbol{w}_k^{\top}\boldsymbol{x}_i)}{\sum_{j=1}^{K}\exp(\boldsymbol{w}_j^{\top}\boldsymbol{x}_i)}\right).$$

The gradient of the loss is expressed as

$$\frac{\partial\mathcal{L}}{\partial\boldsymbol{w}_k} = \frac{1}{M}\sum_{i=1}^{M}\sum_{\ell=1}^{s} y_{i\ell}\frac{\frac{\partial}{\partial\boldsymbol{w}_k}\left[\sum_{j\neq\ell}\exp((\boldsymbol{w}_j - \boldsymbol{w}_\ell)^{\top}\boldsymbol{x}_i)\right]}{1 + \sum_{j\neq\ell}\exp((\boldsymbol{w}_j - \boldsymbol{w}_\ell)^{\top}\boldsymbol{x}_i)},$$

where

$$\frac{\partial}{\partial\boldsymbol{w}_k}\left[\sum_{j\neq\ell}\exp((\boldsymbol{w}_j - \boldsymbol{w}_\ell)^{\top}\boldsymbol{x}_i)\right] = \begin{cases} -\sum_{j\neq k}\exp((\boldsymbol{w}_j - \boldsymbol{w}_k)^{\top}\boldsymbol{x}_i)\boldsymbol{x}_i & \text{if} \quad k = \ell, \\ \exp((\boldsymbol{w}_k - \boldsymbol{w}_\ell)^{\top}\boldsymbol{x}_i) & \text{if} \quad k \neq \ell. \end{cases}$$

Therefore,

$$\frac{\partial\mathcal{L}}{\partial\boldsymbol{w}_k} = \frac{1}{M}\sum_{i=1}^{M}\left\{y_{ik}\frac{-\sum_{j\neq k}\exp((\boldsymbol{w}_j - \boldsymbol{w}_k)^{\top}\boldsymbol{x}_i)}{1 + \sum_{j\neq k}\exp((\boldsymbol{w}_j - \boldsymbol{w}_k)^{\top}\boldsymbol{x}_i)} + \sum_{\ell\neq k} y_{i\ell}\frac{\exp((\boldsymbol{w}_k - \boldsymbol{w}_\ell)^{\top}\boldsymbol{x}_i)}{1 + \sum_{j\neq\ell}\exp((\boldsymbol{w}_j - \boldsymbol{w}_\ell)^{\top}\boldsymbol{x}_i)}\right\}\boldsymbol{x}_i$$

$$= \frac{1}{M}\sum_{i=1}^{M}\left\{\sum_{\ell\neq k} y_{i\ell}\frac{\exp(\boldsymbol{w}_k^{\top}\boldsymbol{x}_i)}{\sum_{j=1}^{s}\exp(\boldsymbol{w}_j^{\top}\boldsymbol{x}_i)} - y_{ik}\frac{\sum_{j\neq k}\exp(\boldsymbol{w}_j^{\top}\boldsymbol{x}_i)}{\sum_{j=1}^{s}\exp(\boldsymbol{w}_j^{\top}\boldsymbol{x}_i)}\right\}\boldsymbol{x}_i$$

$$= \frac{1}{M}\sum_{i=1}^{M}\left\{\sum_{\ell\neq k} y_{i\ell}\frac{\exp(\boldsymbol{w}_k^{\top}\boldsymbol{x}_i)}{\sum_{j=1}^{s}\exp(\boldsymbol{w}_j^{\top}\boldsymbol{x}_i)} - y_{ik}\left(1 - \frac{\exp(\boldsymbol{w}_k^{\top}\boldsymbol{x}_i)}{\sum_{j=1}^{s}\exp(\boldsymbol{w}_j^{\top}\boldsymbol{x}_i)}\right)\right\}\boldsymbol{x}_i$$

$$= \frac{1}{M}\sum_{i=1}^{M}\left\{\underbrace{\sum_{\ell=1}^{s} y_{i\ell}}_{=1}\frac{\exp(\boldsymbol{w}_k^{\top}\boldsymbol{x}_i)}{\sum_{j=1}^{s}\exp(\boldsymbol{w}_j^{\top}\boldsymbol{x}_i)} - y_{ik}\right\}\boldsymbol{x}_i$$

$$= \frac{1}{M}\sum_{i=1}^{M}\left\{\frac{\exp(\boldsymbol{w}_k^{\top}\boldsymbol{x}_i)}{\sum_{j=1}^{s}\exp(\boldsymbol{w}_j^{\top}\boldsymbol{x}_i)} - y_{ik}\right\}\boldsymbol{x}_i$$

Finally, solving for $\frac{\partial\mathcal{L}}{\partial\boldsymbol{w}_k} = \boldsymbol{0}$ yields (2). $\square$

## 8.2 Proofs for the Fully Synthetic Case

*Proof of Theorem 1.* Write $p_i^{(m)} = X_i^{(m)}/n$ where $X_i^{(m)} \sim B(n, p_i^{(m-1)})$ is binomial when conditioned on $\boldsymbol{p}^{(m-1)}$. So we have

$$\mathbb{E}[p_i^{(m)2} \mid \boldsymbol{p}^{(m-1)}] = \frac{1}{n}p_i^{(m-1)} + \left(1 - \frac{1}{n}\right)p_i^{(m-1)2}$$

$$\sum_{i=1}^{s} \mathbb{E}[p_i^{(m)2} \mid \boldsymbol{p}^{(m-1)}] = \frac{1}{n} + \left(1 - \frac{1}{n}\right)\sum_{i=1}^{s} p_i^{(m-1)2}$$

and by the law of total expectation

$$S_m = \frac{1}{n} + \left(1 - \frac{1}{n}\right)S_{m-1} = 1 - \left(1 - \frac{1}{n}\right)^m (1 - S_0). \tag{14}$$

Note that $S_m \nearrow 1$ as $m \to \infty$ and

$$S_m = \mathbb{E}\left[\sum_i p_i^{(m)} p_i^{(m)}\right] \leq \mathbb{E}\left[\left(\sum_i p_i^{(m)}\right)\|\boldsymbol{p}^{(m)}\|_\infty\right] = \mathbb{E}\|\boldsymbol{p}^{(m)}\|_\infty.$$

Recall that $\rho_m := \mathbb{P}\left(\|\boldsymbol{p}^{(m)}\|_\infty = 1\right)$ and $T := \inf\{m \in \mathbb{N} : \|\boldsymbol{p}^{(m)}\|_\infty = 1\} \geq 1$. Since $\sigma_m \geq 1/\tilde{s}$ and $\|\boldsymbol{p}^{(m)}\|_\infty \in \{\frac{1}{n}, \ldots, \frac{n-1}{n}, 1\}$, we have

$$1 \cdot \rho_m + \frac{1}{\tilde{s}}(1 - \rho_m) \leq \mathbb{E}\sigma_m = S_m \leq \mathbb{E}\|\boldsymbol{p}^{(m)}\|_\infty \leq 1 \cdot \rho_m + (1 - 1/n)(1 - \rho_m),$$

from which (7) follows thanks to (14).

For $k = 1, 2, \ldots$ we have $\mathbb{P}(T > k) = \mathbb{P}(\|\boldsymbol{p}^{(k)}\|_\infty < 1) = 1 - \rho_k$ and thus

$$\frac{1 - S_0}{1 - 1/\tilde{s}}(1 - 1/n)^k \leq \mathbb{P}(T > k) \leq n(1 - S_0)(1 - 1/n)^k,$$

which establishes (8) because $\mathbb{E}[T] = \sum_{k=0}^{\infty} \mathbb{P}(T > k)$. □

*Proof of Proposition 1.* Fix $i \in [s]$. For $m \geq 1$ consider the events $E_m := \{T \leq m\}$ and $F_m := \{\boldsymbol{p}_i^{(m)} = 1\}$. Then $\boldsymbol{p}_i^{(m)} \in \{0, 1\}$ on $E_m$ and $F_m \subseteq E_m$. Observe that

$$E_m \nearrow \cup_m E_m \quad \text{and} \quad F_m \nearrow \cup_m F_m = \{\lim_{m \to \infty} \boldsymbol{p}^{(m)} = \delta_i\},$$

where $\mathbb{P}(\cup_m E_m) = 1$ by Theorem 1 and in particular $\mathbb{P}(E_m) > 0$ for $m$ large. Thus

$$p_i = \lim_{m \to \infty} \mathbb{E}[p_i^{(m)}] = \lim_{m \to \infty}\left[\mathbb{P}(E_m)\mathbb{E}[p_i^{(m)} \mid E_m] + \mathbb{P}(E_m^c)\mathbb{E}[p_i^{(m)} \mid E_m^c]\right]$$

$$= \lim_{m \to \infty} \mathbb{E}[p_i^{(m)} \mid E_m] = \lim_{m \to \infty}\mathbb{P}(p_i^{(m)} = 1 \mid E_m) = \lim_{m \to \infty}\frac{\mathbb{P}(F_m \cap E_m)}{\mathbb{P}(E_m)}$$

$$= \lim_{m \to \infty}\frac{\mathbb{P}(F_m)}{\mathbb{P}(E_m)} = \lim_{m \to \infty}\mathbb{P}(F_m) = \mathbb{P}(\cup_m F_m),$$

which proves the assertion. □

## 8.3 Proofs for the Partially Synthetic Case

*Proof of Theorem 2.* Let $v_i^{(m)} := \mathbb{E}\left[p_i^{(m)2}\right]$, $N' := N + n$ and $\boldsymbol{y}_i^{(m)} = (y_{1,i}^{(m)}, \ldots, y_{s,i}^{(m)})$. Then

$$v_i^{(1)} = \frac{p_i}{N} + \left(1 - \frac{1}{N}\right)p_i^2$$

and for $m \geq 2$

$$
\begin{aligned}
v_i^{(m)} &= \frac{1}{N'^2} \mathbb{E} \left[ \left( \sum_{k=1}^{N} y_{i,k}^{(0)} \right)^2 + \left( \sum_{k=1}^{n} y_{i,k}^{(m-1)} \right)^2 + 2 \sum_{j=1}^{N} \sum_{k=1}^{n} y_{i,k}^{(0)} y_{i,k}^{(m-1)} \right] \\
&= \frac{1}{N'^2} \left[ N p_i + (N^2 - N) p_i^2 + n p_i + (n^2 - n) v_i^{(m-1)} \right] + \frac{2}{N'^2} \sum_{j=1}^{N} \sum_{k=1}^{n} \mathbb{E} \left[ y_{i,j}^{(0)} y_{i,k}^{(m-1)} \right] \\
&= \frac{p_i}{N'} + \frac{N^2 - N}{N'^2} p_i^2 + \frac{n^2 - n}{N'^2} v_i^{(m-1)} + \frac{2}{N'^2} \sum_{j=1}^{N} \sum_{k=1}^{n} \mathbb{E} \left[ y_{i,j}^{(0)} y_{i,k}^{(m-1)} \right]. \tag{15}
\end{aligned}
$$

For $m \geq 2$, since $y_{i,k}^{(m-1)}$ is conditionally independent of $y_{i,j}^{(0)}$ given $p_i^{(m-1)}$, by conditioning on $y_{i,j}^{(0)}$ and $p_i^{(m-1)}$ we have

$$
\mathbb{E} \left[ y_{i,j}^{(0)} y_{i,k}^{(m-1)} \right] = \mathbb{E} \left[ y_{i,j}^{(0)} p_i^{(m-1)} \right]
$$

and hence for $m \geq 2$,

$$
v_i^{(m)} = \frac{p_i}{N'} + \frac{N^2 - N}{N'^2} p_i^2 + \frac{n^2 - n}{N'^2} v_i^{(m-1)} + \frac{2n}{N'^2} \sum_{j=1}^{N} \mathbb{E} \left[ y_{i,j}^{(0)} p_i^{(m-1)} \right].
$$

By the definition of $p_i^{(m)}$, for $m \geq 3$

$$
\begin{aligned}
\mathbb{E} \left[ y_{i,j}^{(0)} p_i^{(m-1)} \right] &= \frac{1}{N'} \mathbb{E} \left[ y_{i,j}^{(0)} \sum_{k=1}^{N} y_{i,k}^{(0)} \right] + \frac{1}{N'} \mathbb{E} \left[ y_{i,j}^{(0)} \sum_{k=1}^{n} y_{i,k}^{(m-2)} \right] \\
&= \frac{p_i + (N-1) p_i^2}{N'} + \frac{n}{N'} \mathbb{E} \left[ y_{i,j}^{(0)} p_i^{(m-2)} \right]
\end{aligned}
$$

and

$$
\mathbb{E} \left[ y_{i,j}^{(0)} p_i^{(1)} \right] = \frac{1}{N} p_i + \left( 1 - \frac{1}{N} \right) p_i^2 = \frac{p_i + (N-1) p_i^2}{N}
$$

which gives

$$
\mathbb{E} \left[ y_{i,j}^{(0)} p_i^{(m-1)} \right] = \frac{1 - \left( \frac{n}{N'} \right)^{m-2}}{1 - \frac{n}{N'}} \frac{p_i + (N-1) p_i^2}{N'} + \left( \frac{n}{N'} \right)^{m-2} \mathbb{E} \left[ y_{i,j}^{(0)} p_i^{(1)} \right].
$$

Setting $\alpha := n/N'$, we have $N = (1 - \alpha) N'$ and hence

$$
\mathbb{E} \left[ y_{i,j}^{(0)} p_i^{(m-1)} \right] = \frac{p_i + (N-1) p_i^2}{N}
$$

for all $m \geq 2$. Plugging this back to the expression (15) for $v_i^{(m)}$ gives

$$
v_i^{(m)} = \frac{(1 - \alpha)(1 + 2\alpha)}{N} p_i + \left( 1 - \frac{1}{N} \right) (1 - \alpha^2) p_i^2 + \alpha \left[ \left( 1 + \frac{1}{N} \right) \alpha - \frac{1}{N} \right] v_i^{(m-1)}.
$$

Let $\beta := \alpha \left[ \left( 1 + \frac{1}{N} \right) \alpha - \frac{1}{N} \right]$. Then for $m \geq 1$,

$$
\begin{aligned}
v_i^{(m+1)} &= \frac{1}{N} \left[ \frac{1 - \beta^{m+1}}{1 - \beta} - \alpha(2 - \alpha) \frac{1 - \beta^m}{1 - \beta} \right] p_i \\
&\quad + \left( 1 - \frac{1}{N} \right) \left[ \frac{1 - \beta^{m+1}}{1 - \beta} - \alpha^2 \frac{1 - \beta^m}{1 - \beta} \right] p_i^2 \\
&= \frac{\frac{1}{N} p_i}{1 + (1 + 1/N)\alpha} \left[ 1 + 2\alpha - \left( 1 - \frac{1}{N} \right) \alpha \beta^m \right] \\
&\quad + \frac{(1 - \frac{1}{N}) p_i^2}{1 + (1 + 1/N)\alpha} \left[ 1 + \alpha - \frac{1}{N} \alpha \beta^m \right]
\end{aligned}
$$

and summing across $i \in [s]$ gives the expression for $S_{m+1}$.

Note that $0 < \beta < \alpha < 1$, which gives both an upper bound and a lower bound on $v_i^{(m)}$. In particular, for $m \geq 2$

$$v_i^{(m)} < \frac{1}{1 + (1 + 1/N)\alpha} \left[ \frac{1}{N}(1 + 2\alpha)p_i + \left(1 - \frac{1}{N}\right)(1 + \alpha)p_i^2 \right]$$

and therefore

$$S_m < \frac{1}{1 + (1 + 1/N)\alpha} \left[ \frac{1}{N}(1 + 2\alpha) + \left(1 - \frac{1}{N}\right)(1 + \alpha)S_0 \right]$$

$$= 1 - \frac{\left(1 - \frac{1}{N}\right)(1 + \alpha)(1 - S_0)}{1 + (1 + 1/N)\alpha},$$

where

$$\gamma := \frac{\left(1 - \frac{1}{N}\right)(1 + \alpha)(1 - S_0)}{1 + (1 + 1/N)\alpha} \in \left[ \frac{1 + \alpha}{2 + 3\alpha}(1 - S_0), 1 - S_0 \right]$$

since $0 \leq 1/N \leq 1/2$. □

We state a more general result which implies Theorem 3.

**Theorem 4.** *Let $G_n(s)$ and $\varphi$ be as in Lemma 1. Then for $k \geq 1$ we have*

$$\mathbb{E}\|p^{(k+1)} - p^{(1)}\|_1 < \frac{1}{N}\sqrt{\frac{n\pi}{\varphi(\zeta)}}G_n(s), \tag{16}$$

*where*

$$\zeta = \frac{1}{2} - \left(\frac{1}{2} - 2\mathbb{E}\left[\max_{A \subseteq [s]} p^{(1)}(A)p^{(1)}([s] \setminus A)\right]\right)\left(\frac{N}{N + n}\right)^2.$$

*Proof of Theorem 4.* Since all generations share the same original data source, we can write

$$p^{(k+1)} = \frac{N}{N + n}p^{(1)} + \frac{n}{N + n}\widehat{p^{(k)}}, \quad \text{where} \quad \widehat{p^{(k)}} = \frac{1}{n}\sum_{i=1}^{n} y_i^{(k)} \tag{17}$$

and $\{y_i^{(k)}\}_{i=1,\dots,n}$ are i.i.d. multinomial with parameter $p^{(k)}$ and one trial ($k \geq 1$). This gives

$$p^{(k+1)} - p^{(1)} = \frac{n}{N + n}\left[\widehat{p^{(k)}} - p^{(1)}\right],$$

and applying the triangle inequality yields

$$\|p^{(k+1)} - p^{(1)}\|_1 = \frac{n}{N + n}\left\|\widehat{p^{(k)}} - p^{(1)}\right\|_1 \leq \frac{n}{N + n}\left\|\widehat{p^{(k)}} - p^{(k)}\right\|_1 + \frac{n}{N + n}\left\|p^{(k)} - p^{(1)}\right\|_1.$$

Taking the expectation and solving the recursion gives for $k \geq 1$

$$\mathbb{E}\|p^{(k+1)} - p^{(1)}\|_1 \leq \sum_{j=1}^{k}\left(\frac{n}{N + n}\right)^{k+1-j}\mathbb{E}\left\|\widehat{p^{(j)}} - p^{(j)}\right\|_1. \tag{18}$$

From Lemma 1, we have

$$\mathbb{P}\left(\left\|\widehat{p^{(k)}} - p^{(k)}\right\|_1 \geq t \,\middle|\, p^{(k)}\right) \leq G_n(s)e^{-n\varphi(\pi_k)t^2/4}$$

where $\pi_k := \pi_{p^{(k)}}$. Integrating over $t \in [0, \infty)$, we get

$$\mathbb{E}\left[\left\|\widehat{p^{(k)}} - p^{(k)}\right\|_1 \,\middle|\, p^{(k)}\right] \leq G_n(s)\sqrt{\frac{\pi}{n\varphi(\pi_k)}},$$

and by Jensen's inequality and the concavity of $x \mapsto \varphi(x)^{-1/2}$,

$$\mathbb{E}\left\|\widehat{\boldsymbol{p}^{(k)}} - \boldsymbol{p}^{(k)}\right\|_1 \leq \sqrt{\frac{\pi}{n}} G_n(s) \mathbb{E}\left[\varphi(\pi_k)^{-1/2}\right] \leq \sqrt{\frac{\pi}{n\varphi(\mathbb{E}\pi_k)}} G_n(s).$$

Thus

$$\mathbb{E}\|\boldsymbol{p}^{(k+1)} - \boldsymbol{p}^{(1)}\|_1 \leq \sqrt{\frac{\pi}{n}} G_n(s) \sum_{j=1}^{k} \left(\frac{n}{N+n}\right)^{k+1-j} \frac{1}{\sqrt{\varphi(\mathbb{E}\pi_j)}}. \tag{19}$$

It remains to upper upper bound $\mathbb{E}\pi_j$ since $\varphi$ is decreasing. For $A \subseteq [s]$ let $A^c$ denote its complement. Then by (17), for $A \subseteq [s]$ we have

$$\boldsymbol{p}^{(k+1)}(A)\boldsymbol{p}^{(k+1)}(A^c) = \frac{N^2}{(N+n)^2}\boldsymbol{p}^{(1)}(A)\boldsymbol{p}^{(1)}(A^c) + \frac{n^2}{(N+n)^2}\widehat{\boldsymbol{p}^{(k)}}(A)\widehat{\boldsymbol{p}^{(k)}}(A^c)$$

$$+ \frac{Nn}{(N+n)^2}\left[\boldsymbol{p}^{(1)}(A^c)\widehat{\boldsymbol{p}^{(k)}}(A) + \boldsymbol{p}^{(1)}(A)\widehat{\boldsymbol{p}^{(k)}}(A^c)\right]$$

$$\leq \frac{N^2}{(N+n)^2}\boldsymbol{p}^{(1)}(A)\boldsymbol{p}^{(1)}(A^c) + \frac{n^2}{(N+n)^2}\widehat{\boldsymbol{p}^{(k)}}(A)\widehat{\boldsymbol{p}^{(k)}}(A^c) + \frac{Nn}{(N+n)^2}.$$

Let $\lambda_k := \max_{A \subseteq [s]} \boldsymbol{p}^{(k)}(A)\boldsymbol{p}^{(k)}(A^c)$, so the inequality above implies

$$\mathbb{E}[\lambda_{k+1} \mid \boldsymbol{p}^{(1)}, \boldsymbol{p}^{(k)}] \leq \frac{N^2}{(N+n)^2}\lambda_1 + \frac{n^2}{(N+n)^2}\mathbb{E}\left[\max_{A \subseteq [s]} \widehat{\boldsymbol{p}^{(k)}}(A)\widehat{\boldsymbol{p}^{(k)}}(A^c) \middle| \boldsymbol{p}^{(k)}\right] + \frac{Nn}{(N+n)^2}$$

$$\leq \frac{N^2}{(N+n)^2}\lambda_1 + \frac{n^2}{4(N+n)^2} + \frac{Nn}{(N+n)^2}$$

and thus

$$\mathbb{E}[\lambda_{k+1}] \leq \frac{N^2\mathbb{E}[\lambda_1] + Nn + n^2/4}{(N+n)^2} = \frac{1}{4} - \left(\frac{1}{4} - \mathbb{E}[\lambda_1]\right)\left(\frac{N}{N+n}\right)^2.$$

Observe that for $0 \leq x \leq 1$ we have $\min(x, 1-x) \leq 2x(1-x)$, so $\mathbb{E}[\pi_k] \leq 2\mathbb{E}[\lambda_k]$. Writing

$$\zeta := \frac{1}{2} - \left(\frac{1}{2} - 2\mathbb{E}[\lambda_1]\right)\left(\frac{N}{N+n}\right)^2$$

we have $\mathbb{E}[\pi_k] \leq \zeta$, so from (19) we see that

$$\mathbb{E}\|\boldsymbol{p}^{(k+1)} - \boldsymbol{p}^{(1)}\|_1 \leq \sqrt{\frac{\pi}{n\varphi(\zeta)}} G_n(s) \sum_{j=1}^{k} \left(\frac{n}{N+n}\right)^{k+1-j} < \frac{1}{N}\sqrt{\frac{n\pi}{\varphi(\zeta)}} G_n(s). \tag{20}$$

$\square$

*Proof of Theorem 3.* Theorem 3 is an immediate consequence of Theorem 4 by replacing $\varphi$ with its minimum value 2. $\square$

# 9 Additional Experiments

## 9.1 Architecture & training parameters for GPT2 experiments

We consider that all generation models have GPT2-type[4] vanilla architecture which is a decoder-only generative model with the configuration and training parameters as summarized in Table 1. These parameters were chosen to achieve the best validation loss when training the first-generation model on data produced by ground-truth generative model $\boldsymbol{p}^{(0)}$ as defined in Section 4.

---

[4]https://github.com/karpathy/nanoGPT

| Context length | 128 |
| Embedding dimension | 256 |
| Number of layers | 8 |
| Number of self-attention heads | 4 |
| Vocabulary size | 65 |
| Dropout | 0.2 |
| Learning rate | $10^{-3}$ |
| Batch size | 256 |
| Max iterations | 2000 |

Table 1: Architecture and training parameters in the setting of Section 4.

## 9.2 Additional Experiments with the Statistical Model

To further investigate and demonstrate the generality of our theoretical findings, we present empirical results on two more scenarios that better represent recursive training in real-world settings, as follows:

- *Most Recent Models:* Each generation $p^{(m)}$ is trained on synthetic data from the most recent $K$ models for a fixed window size $K$. More precisely, for some fixed $n \in \mathbb{N}$ we let $n_t^{(m)} = \lfloor n/K \rfloor \cdot \mathbb{1}\{\max(0, m - K) \leq t \leq m - 1\}$ for all $m \geq 1$. When $K = 1$, this case degenerates to the *Fully Synthetic* setting.

- *Randomly Sampled Data:* Each generation $p^{(m)}$ is trained on a mixture of synthetic data from possibly all the previous models and the real data. More precisely, the $m$-th generation model is trained on $n = n_0^{(m)} + \cdots + n_{m-1}^{(m)}$ samples with

$$n_t^{(m)} := \sum_{i=1}^{n} \mathbb{1}\{g_i = t\}, \quad (t = 0, \cdots, m - 1),$$

  samples from the $t$-th generation, where $\{g_i\}_{i \in [n]}$ are independent and uniformly distributed on $\{0, 1, \ldots m - 1\}$ indexing previous-generation models. This setting describes the scenario where data generated by all past models are mixed in a pool from which the training data for the next generation is collected.

Experiments for these two cases are shown in Figure 4. Column 1 corresponds to *Fully Synthetic*, columns 2 and 3 for *Most Recent Models*, and column 4 shows the case of *Randomly Sampled Data*. As we can see, when the window size $K$ is increased, total collapse is delayed but still eventually happens. Observe that there are multiple visible horizontal yellow lines in the second row for window size 1. This is because $p^{(m)}$ can converge to any of the Dirac masses $\delta_i$ provided $p_i > 0$ by Proposition 1. A similar phenomenon can be observed in the second and third columns. On the other hand, in column 4 the model does deteriorate but the deviation plateaus very quickly. In particular, the 100 runs produce only one Dirac mass over the span of 500 generations, suggesting that randomly sampling from all the past models is qualitatively different from sampling from the most recent $K$ models with a fixed $K$. We remark that for $K > 1$, even if $p^{(m)}$ is some Dirac mass, $p^{(m+1)}$ could still regain randomness from models before generation $m$, but with a fixed window size all the randomness eventually disappears.

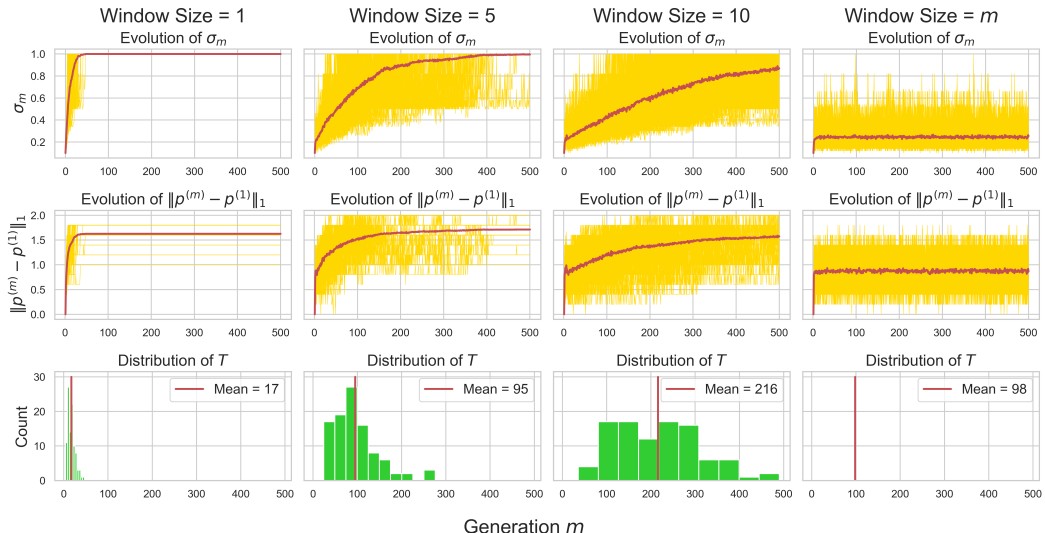

Figure 4: **Experiment for *Most Recent Models* and *Randomly Sampled Data*.** A hundred experiments were run for 500 generations for different window sizes and a fixed sample size $n = 10$. In the first two rows, each yellow line represents the evolution of $\sigma_m$ and $\|p^{(m)} - p^{(1)}\|_1$ in one experiment respectively, with the red line being the empirical mean across 100 runs. The bottom row plots the histogram of total collapse time $T$ with the red line being the empirical mean. The initial distribution $p$ satisfies $s = 600$, $\tilde{s} = 52$ and $S_0 = 0.1$.

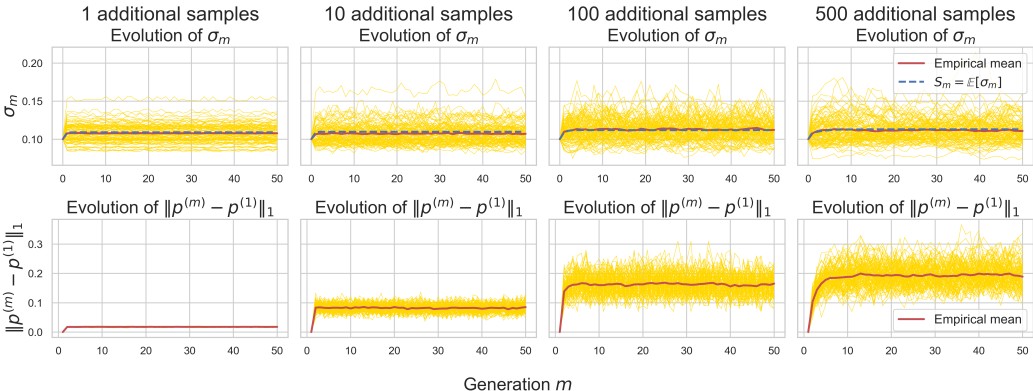

Figure 5: **Partially Synthetic case with different sample sizes $n$.** A hundred experiments were run for 50 generations for $N = 100$ and different values of $n$. Each yellow line represents the evolution of $\sigma_m$ (top row) or $\|p^{(m)} - p^{(1)}\|_1$ (bottom row) in one experiment, with the red line being the empirical mean across 100 runs. The blue dashed lines plot the formula for $S_m$ given by (9). The initial distribution $p$ satisfies $s = 600$, $\tilde{s} = 52$, and $S_0 = 0.1$.

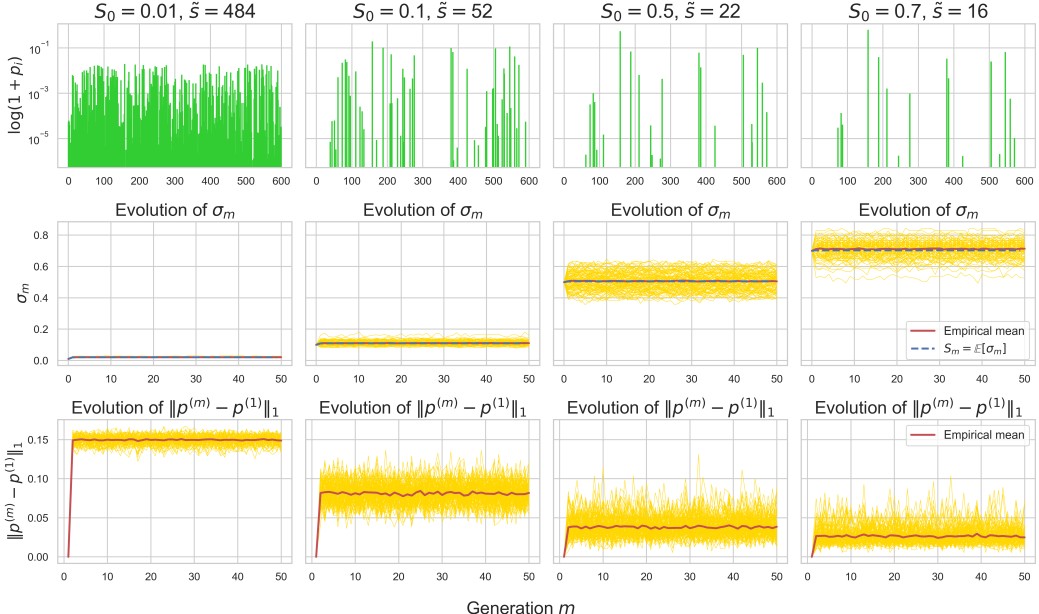

Figure 6: **Partially Synthetic case with different initial distributions $p$.** A hundred experiments were run for 50 generations for $n = 10$, $N = 100$ and different initial distributions $p$ shown in the top row. Notice that as $S_0$ increases, the deviation $\|p^{(m)} - p^{(1)}\|_1$ decreases, as suggest by the inequality (19).

