# OpenReview forum: "How bad is training on synthetic data? A statistical analysis of language model collapse"
_colmweb.org/COLM/2024/Conference — COLM_

### Official Review · Reviewer_Uzr5 · 2024-05-10

**Rating:** 7
**Confidence:** 3
**Ethics Flag:** 1

**Summary:**

This paper presents a theoretical analysis on language model collapse from model generated synthetic data, in a recursive training setup. In this setup, the new generation of model is trained on synthetic data generated from the last generation. From theoretical perspective, the authors prove that model generated synthetic data will cause language model collapse when it is the single data source in the recursive training setup. When combining with real training data, the authors give an upper bound of model generated synthetic data to avoid the collapse. Besides, this paper also provides some empirical results and analysis to align with their theoretical anaylsis.

Overall, the conclusions and analysis presented in this paper are interesting and insightful for language model collapse related research. It is an important step to understand the impact of model generated data in LLM training.

**Reasons To Accept:**

This paper presents a solid theoretical analysis to reveal the impact of model generated synthetic data in recursive training setup. This paper would be interesting to LLM developers and researchers to get them more insights.

**Reasons To Reject:**

Though the analysis is interesting, all the conclusions in this paper have a strong condition: recursive training setup. This condition might not apply to most of the existing LLM effort. Besides, the empirical results in this paper are based on a lightweight setup, while realistic LLM training has much larger scale. It is not clear if the same empirical results can hold on large scale setup.

---

> ### Author Rebuttal · Authors · 2024-05-31
>
> We thank the reviewer for the positive feedback and suggestions. We recognize that our main statistical analysis results are based on the more restrictive recursive training setup and that our experiments are conducted on a small model. Due to limited computational resources, we did not conduct experiments on larger language models. In addition, we address the first limitation in App. C.2 (Fig 4) by performing experiments in more realistic settings “Most Recent Models” and “Randomly Sampled Data”. These two settings are meant to model the scenario of collecting text data online which have been contaminated with machine-generated content. In particular, when data generated by the most recent K models are sampled as the training set (the Most Recent Models setting), model collapse is delayed but still eventually happens, qualitatively similar to the Fully Synthetic setting. On the other hand, our theoretical result suggests that the Partially Synthetic setting behaves differently qualitatively. The deviation probability in Theorem 4 provides guidance on how much real data to incorporate to curb distribution shift.

---

### Official Review · Reviewer_3UBU · 2024-05-10

**Rating:** 6
**Confidence:** 2
**Ethics Flag:** 1

**Summary:**

The paper addresses a problem that has emerged as part of the move towards large-scale generative AI content. More specifically, the contribution is primarily of a theoretical nature but this also gets supported by some corresponding experimental work. The research question is to establish what happens to a model that gets augmented with synthetic data (or is entirely based on synthetic) data when recursively being trained on the output of the next generation of training. The take-home message is that fully synthetic data will ultimately lead to what has been coined “model collapse” in prior work. However, augmenting real data with synthetic data is shown to be less prone to the problem, in particular when the proportion of synthetic data is being controlled.

**Questions To Authors:**

Please see comments above

**Reasons To Accept:**

There are several key strengths of the paper including:

(1) Topically this is a great fit.

(2) The use of synthetic data to increase the pool of training data has become a very active research area, in particular since the release of GPT-2 a few years ago. This could be done for a number of reasons but often due to the lack of quality training data (and sometimes the cost associated with annotating more). My reading of the paper is that the authors foresee a future where much of the data out there is being generated by LLMs (although I do not think this is made explicit in the paper).

(3) The authors provide a detailed theoretical analysis and back up their findings with (some limited) experimental work.

**Reasons To Reject:**

I have a number of concerns, some about the underlying assumptions and others in regards of the experimental work:

(1) The use of synthetic data has been demonstrated to be effective in some settings (and for some NLP tasks) but not for others. However, the usual approach appears to be that of data augmentation where a proportion of synthetic data is being generated (and then filtered before being added to the original dataset). I also understand the appeal of plain synthetic data in circumstances where data privacy might be an issue (e.g. synthetic patient records). However, I wonder how realistic the assumption of continuous recursive training is. Clearly, theoretically it is something worth exploring but I am lacking the actual use case.

(2) In the discussion of related work I would like to hear more about different approaches / paradigms / families of synthetic data generation and then (later) have a discussion about how that choice might affect the findings.

(3) The assumptions made in the theoretical setup (Section 2.1) simplify the concept of language models quite substantially (e.g. not considering transformer architectures). How generalisable are the findings with such simplifying assumptions?

(4) Similarly, the choice of experimental setup in Section 4 is also quite narrow. A fairly tiny corpus is picked on a simple prediction task and GPT-2 is being deployed. All this hints at difficulties of obtaining generalisable insights for practical applications at scale.

(5) There might be some good answers to my concerns and part of this should probably be done as part of a Limitations section. Currently there is no such discussion.


Some other issues:

(1) It would be good to include some compelling running examples. The authors reference (Shumailov et al., 2023), and that paper appears to be doing a great job in that respect.

(2) Almost all entries in the bibliography have essential details missing.

---

> ### Author Rebuttal · Authors · 2024-05-31
>
> We thank the reviewer for their feedback and suggestions. We provide responses to the reviewer's main concerns.
>
> 1. To study *the impact (over multiple generations) of using synthetic data in the pretraining of LLMs*, we consider recursive training where each generation is trained on either synthetic data or a combination of synthetic and real data. The Fully Synthetic setting (while it may not correspond to any practical use case) proves the point that information will eventually be lost over generations if we train new LLMs solely on synthetic data. On the contrary, in The Partially Synthetic setting (real+synthetic), degradation in model performance can be controlled if we carefully choose the ratio of real to synhetic data
>
> 2. Synthetic data can indeed be generated and used in various ways. E.g, one could *prune* the synthetic dataset to only keep low probability samples (to make them more present in the training data for next gen LLMs). In this paper, we only consider synthetic data obtained directly by sampling the output of an LLM (without post-sampling processing). Other approaches are beyond the scope of this work. We will include this discussion in the revised paper.
>
> 3. Our theoretical results consider a simple model for tractability. Extending the theory to transformers is currently challenging given the fact that the transformer architecture is highly complex. However, our experiments demonstrate the generalizability of our findings (derived on the simple model) in the context of transformers. In Fig 3, the Fully Synthetic and Partially Synthetic settings behave qualitatively differently – the former exhibits model collapse while the latter stabilizes in terms of model deviation. Even though the exact quantification of distribution drift is difficult to estimate for transformers, the result of this experiment aligns with our theoretical derivations in Theorems 1 and 4.
>
> 4. We respectfully disagree with the reviewer on this point. We do not provide experiments with larger models because it is computationally infeasible given our compute budget. In this regard, we would like to refer the reviewer to the last point of the Review Guidelines, where it is stated that "**we ask that you take into account that most researchers do not have access to large-scale compute**".
>
> 5. Extended limitations section: we will include the aforementioned points and concerns raised by the reviewers.
>
> We hope our response clarifies some potential misunderstandings.

---

> > ### Comment · Reviewer_3UBU · 2024-06-05
> > **Response to rebuttal**
> >
> > I thank the authors for their response to my concerns. The rebuttal does contain partial answers to some of my concerns and I assume that the authors will update the paper accordingly if invited to submit a revision.
> >
> > My two concerns regarding (a) tiny sample size and (b) lack of realistic use case remain as before. I also note that these two points have been raised in the other reviews as well. Nevertheless I will raise my overall score slightly as I do not think that there is a systematic flaw in the work. It simply is not strong enough in my view to warrant a clear accept.

---

> ### Author Response · Authors · 2024-06-04
> **Update**
>
> Given that the discussion deadline is in two days, we would appreciate it if the reviewer could provide feedback on our rebuttal, particularly noting if there is anything unclear.

---

### Official Review · Reviewer_k9FW · 2024-05-13

**Rating:** 6
**Confidence:** 3
**Ethics Flag:** 1

**Summary:**

The paper performs a theoretical analysis of recursive training loops in language model training, i.e., where synthetic data generated by language models contaminates the data used for training future iterations of language models. Prior work has found that this phenomenon leads to "model collapse," where models' outputs become very limited as the amount of synthetic data in the training set increases. This paper builds on one of these works (Shumailov et. al. 2023), particularly exploring the setting of models for next token prediction. They conclude that theoretically, model collapse should consistently occur in the fully synthetic scenario. In the partially synthetic scenario, an upper bound on the proportion of synthetically generated data in the training set is identified to mitigate model collapse. The authors confirm these results using experiments with toy models.

**Questions To Authors:**

* When commenting on the issue that it “is impossible that the learned model captures all the information about the original distribution”, is the issue that results from this related to the capacity of the model being trained, or to the data generated from previous models?
* By canonical vectors, do you mean canonical basis vectors? They’re called one hot embeddings elsewhere, and it may be more clear to stick to one term.
* After some thought, I can see why all models p^{(m)} are unbiased, but I think it would be best to explain this more explicitly to the reader

**Reasons To Accept:**

* The topic of the paper is very relevant, since the amount of model generated data on the web has drastically increased in recent years. Previous studies have explored this phenomenon empirically, but few have provided theoretical insights, which are important when deciding how to build the next generation of of language models.
* The derivations seem sound, although there is a chance that I missed an error in one or more of the proofs.
* There is a comprehensive discussion of related work, giving a nice overview of the recent study of model collapse
* The authors generally provide good intuition for the mathematical results

**Reasons To Reject:**

* The presentation could generally use some work. The paper is dense and hard to follow. For example, in the setup in 2.1, there’s switching between terms, e.g., x_l and e_j which are multiplied in the final linear layer, and N and M for number of training samples. I can’t follow the reasoning for the use of these multiple terms, and they make section 2.1 hard to understand.
* The theoretical analysis is only for a single context. It’s unclear which conclusions would change when considering the large number of possible contexts. While the authors point this shortcoming out, the absence of such an analysis makes the paper much weaker, in my opinion, since this is the main contribution.
* Experimental results are only for toy settings, although it would be difficult to perform experiments on realistically-sized language models with realistically-sized datasets
* The contributions above Shumailov et. al. 2023 are arguably limited

---

> ### Author Rebuttal · Authors · 2024-05-31
>
> We thank the reviewer for the constructive feedback and the recognition that our work is among the first to provide direct theoretical insights into recursively training language models. We would like to bring the following to the reviewer’s attention:
> 1. We choose to give a concise presentation in Sect 2.1 due to space constraint. The distinction between $x_l$ and $e_j$ arises from the assumption that contexts are represented by canonical (basis) vectors $e_j$ (i.e. one-hot embeddings in ML language), as explained in Sect 2.1 (“Note that in current …”). Additionally, in Eq (1), $M$ denotes the number of all (context, next-token) pairs in the dataset, while $N=|C_j|$ denotes the number of data points with context $j$ so $N=N_j$ in fact depends on $j$ and $M=\sum_j N_j$. We suppress the subscript of $N_j$ for brevity. A derivation for $p^{(m)}$ being unbiased will be added to the appendix.
>
> 2. We remark that our main results (in particular Theorems 1 and 4) hold context-wise and easily extend to all contexts under certain conditions. For example, if we assume that the number of synthetic samples $n_j$ for each context j is fixed across generations, we directly obtain uniform bounds on the expected collapse time or distribution shift which holds true for all contexts.
> A natural way to relax this assumption is to consider the autoregressive model governed by Eq (5) so that $n_j$ depends on the conditional distributions given other contexts, but this significantly complicates the analysis. Instead, in Sect 4 we perform exactly such an experiment with transformer model. In the top-left corner of Fig 3, we report the average deviation from the original distribution (across 300 randomly sampled contexts) which confirms the dichotomy between whether or not original data is retained for training, in accordance with our theoretical findings.
>
> 3. Our contributions beyond Shumailov et al. (2023) consist in the direct quantification of LM deterioration in recursive training. Their treatment on LM is indirect: theoretically, they only analyze Gaussian distributions which do not apply to LMs where output distributions are discrete. In contrast, our setup is more closely related to the actual setting in LM recursive training and we provide direct analysis accordingly.

---

> > ### Comment · Reviewer_k9FW · 2024-06-05
> > **Response**
> >
> > Thank you for the clarifications. While some of my concerns are assuaged, my overall perception of the work remains the same. I will therefore keep my score the same.

---

### Decision · Program_Chairs · 2024-07-10

**Decision:**

Accept

**Comment:**

This paper presents a statistical model to characterize the impact of recursive training scenarios to understand model collapse in language models. Reviewers all agreed that this work is very timely relevant and presents a solid theoretical analysis, together with experimental work. There are a few concerns about the assumption of continuous recursive training, the very lightweighted empirical results, as well as generalization to more realistic settings. We suggest the authors to follow reviewers’ comments to revise their draft and also add limitation discussion.

[comments from the PCs] Please follow up on the AC request, and also include a limitation section.